# Laboratory evolution of synthetic electron transport system variants reveals a larger metabolic respiratory system and its plasticity

Amitesh Anand[1,2✉], Arjun Patel[1], Ke Chen [1], Connor A. Olson[1], Patrick V. Phaneuf [1], Cameron Lamoureux[1], Ying Hefner[1], Richard Szubin[1], Adam M. Feist [1,3] & Bernhard O. Palsson [1,3✉]

The bacterial respiratory electron transport system (ETS) is branched to allow condition-specific modulation of energy metabolism. There is a detailed understanding of the structural and biochemical features of respiratory enzymes; however, a holistic examination of the system and its plasticity is lacking. Here we generate four strains of *Escherichia coli* harboring unbranched ETS that pump 1, 2, 3, or 4 proton(s) per electron and characterized them using a combination of synergistic methods (adaptive laboratory evolution, multi-omic analyses, and computation of proteome allocation). We report that: (a) all four ETS variants evolve to a similar optimized growth rate, and (b) the laboratory evolutions generate specific rewiring of major energy-generating pathways, coupled to the ETS, to optimize ATP production capability. We thus define an Aero-Type System (ATS), which is a generalization of the aerobic bioenergetics and is a metabolic systems biology description of respiration and its inherent plasticity.

[1] Department of Bioengineering, University of California, San Diego, La Jolla, CA, USA. [2] Department of Biological Sciences, Tata Institute of Fundamental Research, Mumbai, Maharashtra, India. [3] Novo Nordisk Foundation Center for Biosustainability, Technical University of Denmark, Kemitorvet, Kongens, Lyngby, Denmark. ✉email: amitesh.anand@tifr.res.in; palsson@ucsd.edu

Respiration requires organisms to have an electron transport system (ETS) for the generation of proton-motive force across the membrane that drives ATP synthase. Although the molecular details of the ETS are well-studied and constitute textbook material, few studies have appeared to elucidate its systems biology. The most thermodynamically efficient ETS consists of two enzymes, an NADH: quinone oxidoreductase (NqRED) and a dioxygen reductase ($O_2$RED), which facilitate the shuttling of electrons from NADH to oxygen. However, evolution has produced variations within the ETS which modulate the overall energy efficiency of the system even within the same organism[1–3]. The systems level impact of these variations and their individual physiological optimality remain poorly determined. To mimic varying ETS efficiency we generated four *Escherichia coli* deletion strains (named ETS-1H, 2H, 3H, and 4H), each with one of the four unbranched ETS variants that pump 1, 2, 3, or 4 proton(s) per electron, respectively. We then performed systems level characterization of these ETS variants. We observe that: (a) adaptive laboratory evolution (ALE) enables all four ETS variants to evolve to a similar growth rate; (b) the evolution of ETS variants is supported by specific rewiring of major energy-generating pathways that couple to the ETS to optimize their ATP production capability; (c) proteome allocation per ATP generated is the same for all the variants, (d) the aerotype, that designates the overall ATP generation strategy[4] of a variant, remain conserved during its laboratory evolution, with the exception of the ETS-4H variant; and (e) integrated computational analysis of the data supports a proton-to-ATP ratio of 10 protons per 3 ATP for ATP synthase for all four ETS variants.

## Results and discussion

*E. coli* has a highly flexible ETS consisting of 15 dehydrogenases and 10 reductases to allow growth in both oxic and anoxic environments[5]. The expression of these enzymes is regulated by a variety of electron acceptors with a known hierarchy, such that oxygen represses all anoxic respiratory pathways and nitrate represses other anoxic pathways[3,5]. Despite this thermodynamic hierarchy, co-expression of different respiratory chains was reported in another γ-proteobacterium to expand the flexibility of its electron transfer network[3]. We probed the condition-dependent expression of all these dehydrogenases and reductases using a large RNA-seq compendium for *E. coli*[6]. We observed a spectrum of expression values of these genes across the experimental conditions showing the contribution of these enzymes in generating plasticity in energy metabolism (Supplementary Fig. 1).

To examine the contributions of individual oxic respiratory pathways to bioenergetics, we sought to design unbranched pathways through the ETS. The oxic component is contributed by both proton pumping and non-pumping NqREDs (hereafter referred to as NDH-I and NDH-II, respectively) along with three types of $O_2$REDs (Fig. 1a). Cytochrome bd $O_2$REDs (CBDs) are less electrogenic compared to Cytochrome bo₃ $O_2$REDs (CYO). There are two CBDs, bd-I and bd-II, and both functions similarly to generate proton-motive force (PMF) by a vectorial movement of protons involving transmembrane charge separation. The similar PMF generation strategies make bd-I and bd-II $O_2$REDs equivalent when choosing gene knockout strategies[7,8].

Based on these characteristics, we designed four ETS variants with unbranched electron flows representing all alternate oxic respiratory routes translocating 1, 2, 3, or 4 proton(s) per electron (designated as ETS-nH, with $n = 1, 2, 3, 4$). The designs of the four ETS variants are illustrated in Fig. 1b.

Next, we analyzed their growth phenotype (Fig. 1c). Interestingly, the unevolved variants (called *u*ETS) showed different growth rates that had no clear association with their $H^+/e^-$ value. While the loss of activity of NDH-I showed a lesser growth rate retardation, the deletion of NDH-II significantly compromised the growth rate of the deletion strains.

To allow the ETS variants to overcome the growth defects resulting from gene deletions, we performed ALE with four independent replicates of each variant in an oxic environment (Supplementary Table 1) (evolved variants are named *e*ETS-nHm with the replicate evolutionary endpoints indexed as m = A, B, C, D)[9]. We evolved all variants until their growth rate plateaued. ETS-1H, 2H & 3H required evolution for approximately 400 generations, while ETS-4H required approximately 700 generations. In spite of the different number of protons pumped per electron, all four ETS variants evolved to a similar optimized growth rate in replicate evolutions (~0.85 h⁻¹) (Fig. 1c).

Next, we sought to determine the acquired mutations that enabled adaptation to a higher growth rate for all ETS variants. We performed whole-genome sequencing of each strain and used a comprehensive database of mutations from ALE experiments (aledb.org[10]) to interpret the potential impact of the identified mutations. The mutation calling revealed only a few genetic changes in the evolved strains except for *e*ETS-4HC which acquired 15 genetic changes (Supplementary Data 1). The higher number of mutations in *e*ETS-4HC could be due to the mutated DNA mismatch repair enzyme *mutS* in this strain[11]. Every ETS variant acquired mutations responsible for enabling faster growth on M9 minimal medium (Supplementary Table 2, Supplementary Data 1)[12–15]. An intergenic mutation between *pyrE* and *rph* has been reported to alleviate pyrimidine pseudo-auxotrophy resulting in a faster growth rate. RNA polymerase subunit mutations are proposed to favor a higher growth rate by accelerating the transcriptional processes. Another common mutation reported to support a faster growth rate is in the intergenic region between *hns* and *tdk*. This mutation is expected to downregulate several stress response pathways and shift resources to support growth. *u*ETS-1H carried the *pyrE-rph* intergenic mutation, which explains the relatively faster initial growth rate of this strain.

Besides mutations responsible for acclimatization to media, *u*ETS-3H and *u*ETS-4H acquired a common gene-related mutation in all four independently evolved lineages. This mutational convergence simplified the otherwise difficult task of establishing the genotype-phenotype relationship[16–18].

All four evolved replicates of *u*ETS-3H acquired point mutations in *sdhA*, the catalytic subunit of succinate dehydrogenase (Supplementary Table 2). *e*ETS-3HB acquired a point mutation that brings in a premature termination codon in the *sdhA* open reading frame, suggesting a loss of functional enzyme (Supplementary Data 1). We explored the potential impact of other mutations by investigating whether the SNPs could affect the protein's function based on amino acid properties and sequence homology (SIFT)[19] or structural stability (ΔΔG)[20]. Almost all mutations in *sdhA* were either in or near interface surfaces and seem to be working to disrupt its functionality by either disrupting a substrate-binding site or causing a structural-functional perturbation (Fig. 1d, e). Notably, the deletion of another subunit of this enzyme, *sdhC*, has been reported to increase the biomass yield in an oxic environment[21]. *u*ETS-3H appeared to adopt a similar metabolic route to increase its growth rate.

All four replicates of *u*ETS-4H acquired mutations in an inadequately characterized gene, *yjjX* (Supplementary Table 2, Supplementary Data 1). The structural and biochemical evidence suggests that YjjX, an inosine/xanthosine triphosphatase, may be involved in the mitigation of the deleterious impact of oxidative stress by preventing the accumulation of altered nucleotides[22]. Also, the physical association of YjjX with the elongation factor suggests a negative impact on the translational rate. The

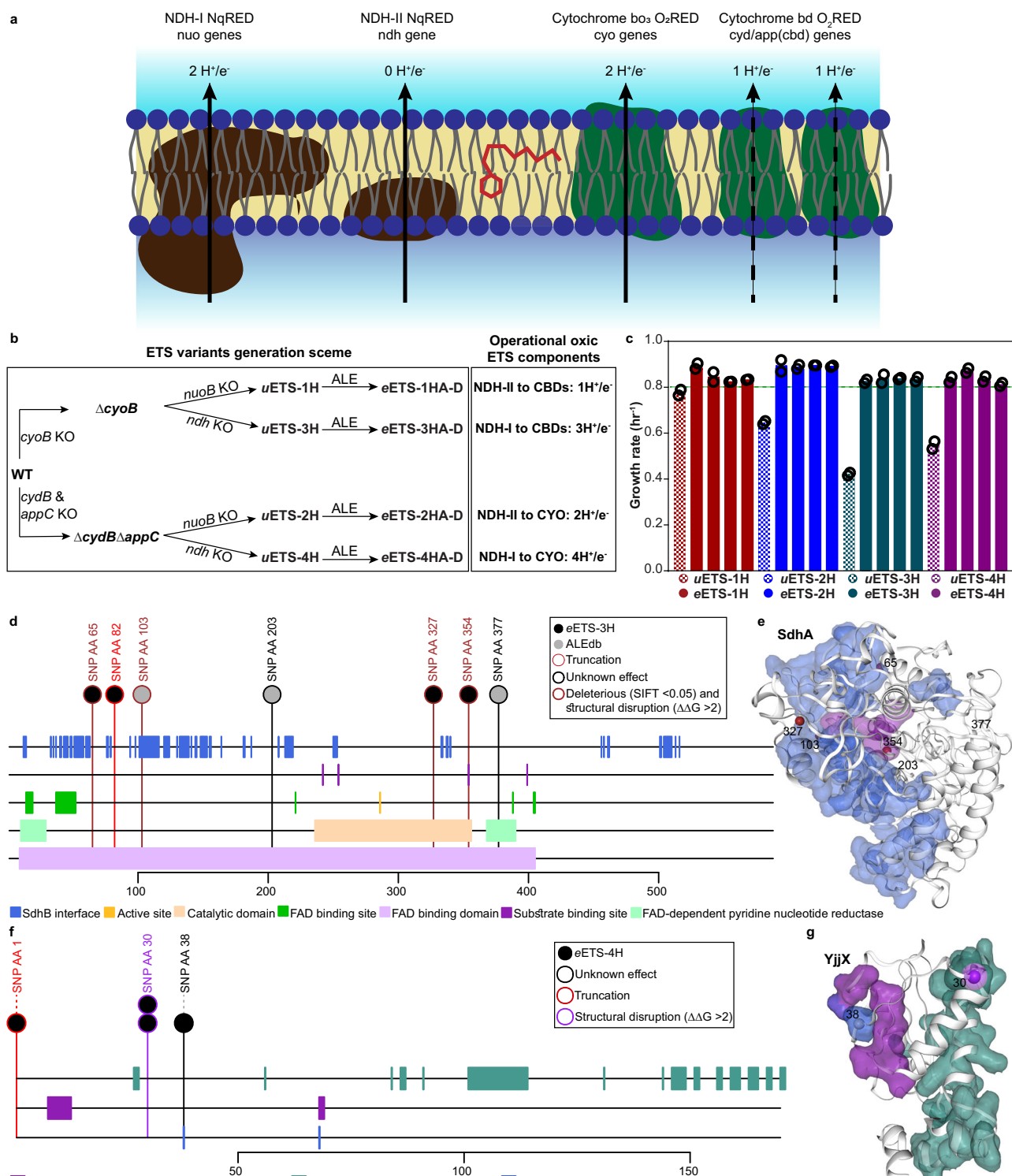

**Fig. 1 Generation and evolution of unbranched ETS variants. a** Schematic showing the respiratory enzymes involved in the flow of electrons from NADH (donor) to oxygen (acceptor). NDH-I and NDH-II are the proton pumping and non-pumping NADH: quinone oxidoreductase, respectively. Dashed arrows for CBDs represent the vectorial mode of PMF generation. **b** Scheme for generating ETS variants translocating 1, 2, 3, or 4 proton(s) per electron. *u*ETS is the unevolved strain and *e*ETS is the evolved strain. A–D are the four independently evolved lineages of each strain. **c** Growth rates of ETS variants before and after ALE. **d**, **f** Predictive mechanistic interpretation of the impact of mutations observed in the evolved strains of (**d**) ETS-3H (*sdhA*) and (**f**) ETS-4H (*yjjX*) mutations on the structure and function of the protein. Mutations displayed are those from this study and other ALE experiments in ALEdb seen to mutate these genes[10]. Horizontal tracks display the reported features associated with the region of the protein. The mutations collected from ALEdb refer to experiments from the following set, respectively[12,49,50]. Protein structures showing the amino acid residues mutated in (**e**) SdhA and (**g**) YjjX. Source Data available in Supplementary Table 3.

STRING-based protein-protein interaction predicted the association of YjjX with glycolytic and ATP biosynthetic processes[23]. Interestingly, *e*ETS-4HA replaced the start codon, ATG, with ATA (Supplementary Data 1). Apart from replacing methionine with isoleucine, this substitution potentially diminishes the expression of this protein[24]. A similar disruptive impact is expected from other *yjjX* mutations (Fig. 1f, g). The cysteine to tyrosine substitution at amino acid residue 30 was predicted to destabilize the structure as it lies just beside a subunit interface residue, and a charge reversal due to the glutamate to lysine substitution at amino acid residue 38 targets the metal-binding site. Thus, it appears that *e*ETS-4H is attempting to prevent translational halting to achieve a higher growth rate.

Since the restoration of the evolved variants to the same growth rate cannot be deciphered from genetic changes alone, we took a broader systems view to understand the underlying metabolic perturbations. We examined how the evolved variants rewired the fluxes through the major metabolic pathways that couple to the ETS. We generated RNA sequencing and metabolite profiling data for all the strains and performed targeted and systems level analyses. We observed a high transcriptional correlation among the evolved replicates (Spearman's rank correlation coefficient >0.75) of each variant, but the correlation between pre-and post-evolved variants was lower (Fig. 2a). Notably, consistent genetic

and transcriptomic changes supported a common evolutionary trajectory for the replicates of each variant.

Bacterial physiology displays a remarkable compensatory potential facilitated by altered metabolic flux states resulting from genetic and transcriptomic changes[21]. Therefore, we examined if the surrogate NqRED or $O_2$RED compensated for the loss of function resulting from deleted ETS enzymes (Fig. 2b). There was no clear compensatory trend in the strains with unbranched ETS except for ETS-4H. ETS-4H increased the expression of NDH-I while increasing or maintaining the expression of CYO after evolution. Surprisingly, the compensatory upregulation of *ndh* in *u*ETS-2H was lost after evolution to a higher growth rate.

Since RNA expression levels may not correlate with metabolic fluxes due to differential translation efficiency and different enzyme catalytic turnover rates, we performed a metabolic flux distribution analysis. To obtain the metabolic flux map, we measured the medium exchange rates of the major metabolites related to respiratory metabolism (Supplementary Table 3). We used both the metabolite exchange rates and transcriptomic data as constraints to simulate the flux through the pathways of the central carbon metabolism using a genome-scale model of metabolism and protein expression (ME-model)[25]. We observed a high correlation in the metabolic flux distributions of the four evolved replicates of each strain, further supporting a similar

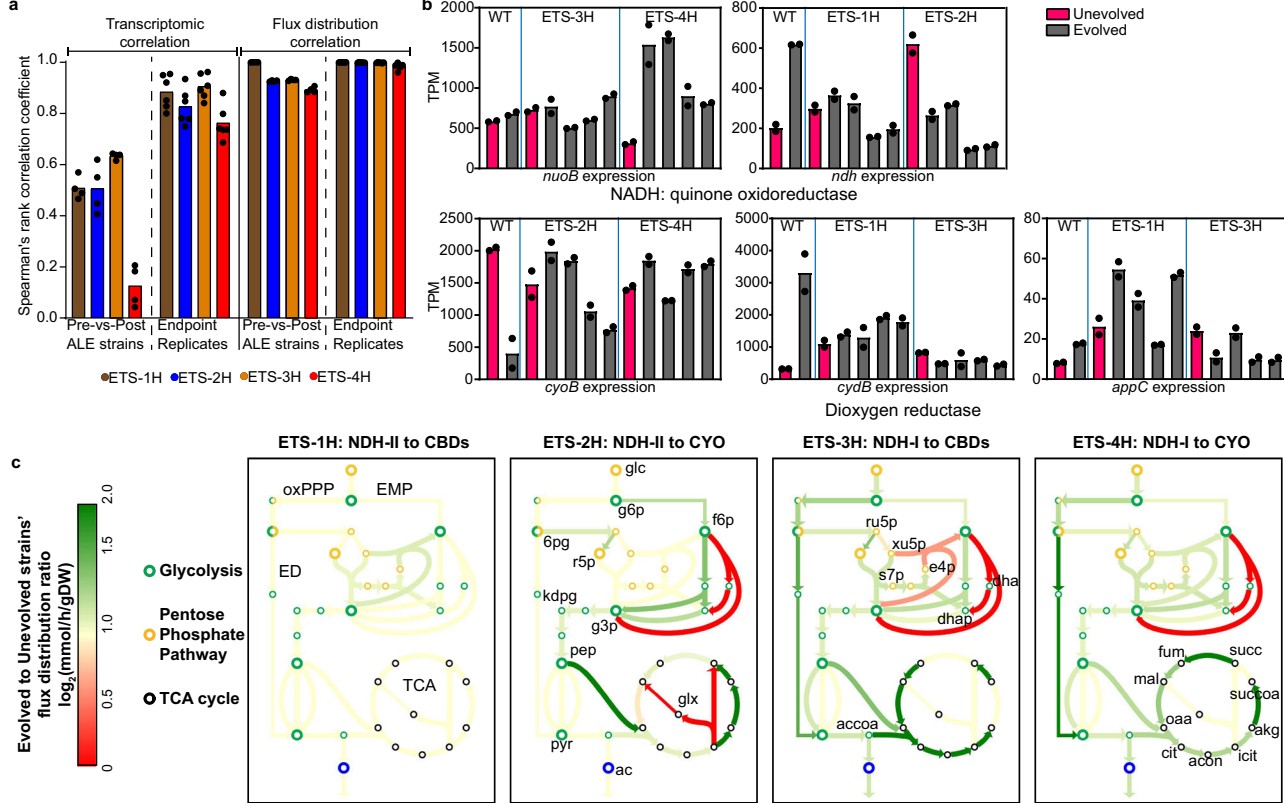

**Fig. 2 Metabolic rewiring supporting growth rate optimization in the ETS variants. a** Transcriptional and metabolic flux distribution correlations between evolved replicates and unevolved strains. Correlation of all four evolved replicates of each ETS variant has been calculated with corresponding unevolved strain and has been presented as Pre-vs.-Post ALE strains correlation. Correlation of all four evolved replicates of each ETS variant has been calculated among each other and has been presented as Endpoint Replicates correlation. **b** Expression changes in the alternate NqRED and $O_2$RED in the unbranched ETS variants. **c** Computed metabolic flux maps depicting the central metabolism in the evolved ETS variants as compared to respective unevolved ETS variants. Key metabolites are indicated in the figure as follows: glc glucose, g6p D-glucose-6-phosphate, f6p D-fructose-6-phosphate, 6 pg 6-phospho-D-gluconate, ru5p D-ribulose-5-phosphate, r5p α-D-ribose 5-phosphate, xu5p D-xylulose-5-phosphate, s7p sedoheptulose-7-phosphate, e4p D-erythrose-4-phosphate, dha dihydroxyacetone, dhap dihydroxyacetone-phosphate, kdpg 2-keto-3-deoxy-6-phosphogluconate, g3p glyceraldehyde-3-phosphate, pep phosphoenolpyruvate, pyr pyruvate, ac acetate, accoa acetyl-CoA, oaa oxaloacetate, cit citrate, acon cis-aconitate, icit isocitrate, akg 2-oxoglutarate, succoa succinyl-CoA, succ succinate, fum fumarate, mal malate. [oxPPP oxidative pentose phosphate pathway, EMP Embden-Meyerhof-Parnas pathway, ED Entner–Doudoroff pathway, TCA Tricarboxylic acid cycle]. Source Data available in Supplementary Table 2 and provided RNA-seq data.

evolutionary pathway followed by replicates of each variant (Fig. 2a).

To more deeply understand the different metabolic states exhibited by the evolved variants, we examined the variations in their computed proteome allocation using the solutions from the phenotypic and transcriptomic constrained ME-models. We observed a clear distinction between strains with alternate NqRED for the preferred glycolytic pathway (Fig. 2c). NDH-I has approximately 10-times higher molecular mass as compared to NDH-II[26,27]. Therefore, despite its PMF generation potential, NDH-I is a less preferred dehydrogenase during oxic respiration to achieve faster growth[5]. The non-proton pumping high turn-over dehydrogenase, NDH-II, is better suited to relieve the growth bottleneck that may arise due to excess built-up of PMF while allowing the operation of oxic ETS[5,28].

The finite resource carrying capacity of a cell creates metabolic tradeoffs on how to partition the proteome to support metabolic pathways best suited for a given growth condition. With an approximately 3.5-fold higher protein cost, the Embden–Meyerhoff–Parnass (EMP) pathway consumes a larger proportion of proteome as compared to the Entner–Doudoroff (ED) pathway[29]. However, the higher ATP yield of the EMP pathway alludes to a potential tradeoff between the two glycolytic pathways for optimizing ATP production while maintaining a growth-supporting proteome[30]. The ETS-3H and ETS-4H strains forced to respire using larger NqRED (NDH-I) increased the flux through the proteome conservative ED pathway. Thus, we observed a compensatory selection of the preferred pathway to achieve a balanced proteome.

Interestingly, while strains with nuoB deletion (ETS-1H and ETS-2H) increased metabolic flux through complex II of ETS, ETS-3H appeared to minimize the flux through complex II (Supplementary Data 2). Notably, eETS-3H lacks ndh and acquired a mutation in the gene sdhA which codes for a complex II subunit. However, ETS-4H, which also lacks the ndh gene, increased the flux through complex II, albeit at a lower level compared to ETS-1H and ETS-2H.

Thus, metabolic plasticity (reflected in metabolic rewiring and associated proteome allocation) allows for redundancy in the eETS variants while supporting the same growth rate. Knowledge of this metabolic plasticity motivated the examination of the overall bioenergetics-state of the evolved ETS variants to fully understand the basis for the evolution to the same growth rate. We have earlier defined an approach to classify the E. coli phenotypes into aero-types, which is a quantitative fitness descriptor based on cellular respiratory behavior and proteome allocation[4]. The stratification of aero-types is based on the multimodal distribution of the fraction of total ATP produced through ATP synthase which is modulated through the discrete usage of ETS enzymes. We have reported a non-uniform distribution of phenotypic growth data in the rate-yield plane that can be approximately segregated in different aero-types based on sampling simulations. Here we used aero-types to examine the fitness distribution of ETS variants.

We observed that ETS-1H, ETS-2H, and ETS-3H did not show a major shift in their biomass yield during evolution and thus preserved their respective aero-types (Fig. 3a). The evolutionary optimization of growth rate appears to be largely driven by rewiring central carbon metabolism while oxidative energy metabolism is conserved. ETS-4H jumped from a lower to a higher aero-type after evolution, suggesting an increase in oxic metabolism. The ETS-4H variant has the highest PMF generation capacity. Its aero-type shift to higher classes occurred only after adaptive evolution.

The clustering of each evolved ETS variant along the same growth rate isocline (Fig. 3a) indicated global remodeling of the energy metabolic network to produce similar growth-supporting bioenergetics. We thus defined a larger respiratory system, called the Aero-Type System (ATS), consisting of oxidative phosphorylation, glycolysis, pyruvate metabolism, the TCA cycle, and the Pentose Phosphate pathway, that together define the overall state of oxic energy metabolism (Supplementary Fig. 3). The total proteome allocated to the ATS was very similar in each eETS variant, and the total ATP output of each proteome expressed was almost constant (Fig. 3b). Thus, the composition of the ATS was malleable and able to provide the same supply of ATP, allowing similar growth rates for all eETS variants. We also observed a trend in the metabolic location of ATP production across the variants, where the relative contribution of oxidative phosphorylation was highest for eETS-4H and lowest for eETS-1H (Fig. 3c). Accordingly, an inverse trend was observed for glycolytic and fermentative ATP production.

We next examined the transcriptome to identify the tradeoffs in gene expression that enabled the different metabolic states. We applied a blind source signal separation algorithm, called independent component analysis (ICA)[31], to examine differential partitioning of the transcriptome of the 209 ATS genes. ICA decomposed the ATS transcriptome into independently modulated sets of genes (called iModulons) (Supplementary Data 3). The activities of several iModulons showed a clear association with the aero-type of the ETS variants (Supplementary Fig. 4). iModulons consisting of genes associated with oxic respiration showed a positive correlation with aero-type status (iModulons 8, 13, and b2287), and those constituted by anoxic and/or metabolic genes showed a negative correlation (iModulons 7, 9, 10, 16, and b3366) (Supplementary Fig. 2). Thus, an oxic-anoxic transcriptomic tradeoff enabled the four ETS variants to maintain similar ATP production capacity (Fig. 3d).

The direct measurement of the number of protons translocated through ATP synthase to produce one molecule of ATP ($H^+$/ATP) is technically challenging and, therefore, it is still an area of active research[32]. The rotational catalysis-based calculation suggests the $H^+$/ATP value to be 3.3, due to the symmetry mismatch between the $F_o$ and $F_1$ complexes of ATP synthase: threefold symmetry of α3β3 in F1 and tenfold symmetry of the c-ring in $F_o$[33,34]. The proton-to-ATP ratio may vary depending upon any change in the number of c-subunits and this modulation allows tailoring to meet the bioenergetic demand of various organisms[35]. The $H^+$/ATP value derived using a synthetically reconstituted membrane system was found to be 4[36]. With our comprehensive definition of the state of the ATS amongst the variants, we could address the issue of ATP synthase proton-to-ATP ratio. We used data generated on the variants to computationally estimate the most likely proton-to-ATP ratio for E. coli ATP synthase[32]. We constrained the ME-model using the observed metabolic exchange rates and gene expression data and optimized for the $H^+$/ATP value of ATP synthase that produces the experimentally estimated growth rates of the variants. The ME-model calculates the median value of the $H^+$/ATP to be 3.25, a value close to 3.3 supporting the rotational catalysis hypothesis (Fig. 3e). Notably, while 10 is the preferred number of c subunits in the E. coli $F_o$ motor of ATP synthase, the number of subunits can vary, which will change the $H^+$/ATP value[37–40].

Taken together, our results lead to an expanded definition of oxic respiration beyond the conventional ETS, which involves an electron transport chain to create PMF, that then drives the ATP synthase. Here, we define the Aero-Type System that encompasses the ETS and coupled metabolic pathways (Supplementary Fig. 3). The ATS is composed of 209 genes (Supplementary Data 3). The ATS represents about 38% proteome allocation in all evolved variants. A decrease in the ETS energetic efficiency (often measured in terms of the P/O ratio) can be balanced by increased

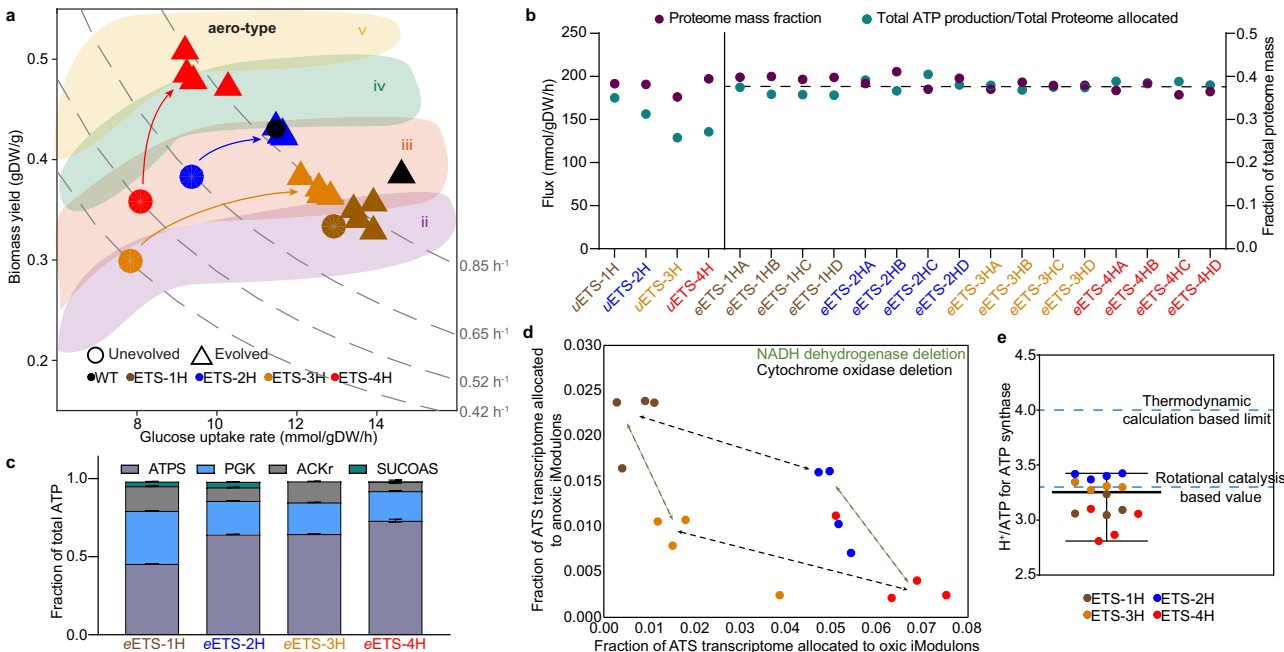

**Fig. 3 Systems-level examination of ETS variants. a** Aero-type classification of the ETS variants. Broken lines on the aero-type plot show growth rate isoclines. **b** ME-model-based examination of the ATP production (left y-axis) and proteome allocation (right y-axis) in the ETS variants. The ATP produced per ATS proteome is approximately the same. **c** Contributions of different ATP-producing reactions towards total ATP production. ATP production by (i) ATP synthase (ATPS) in oxidative phosphorylation, (ii) acetate kinase (ACKr) in mixed acid fermentation, (iii) succinyl-CoA synthetase (SUCOAS) in the TCA cycle, and (iv) phosphoglycerate kinase (PGK) in the glycolysis pathway is shown in the histogram. The mean and standard deviation values are calculated using four independently evolved replicates of each strain. **d** Tradeoffs in the expression levels of genes of iModulons associated with anoxic (y-axis) and oxic (x-axis) energetics underlie the rewiring of the ATS to allow all variants to achieve approximately the same growth rate. The lowest aerotype (ETS-1H) has high anoxic/low oxic gene expression while the highest aerotype (ETS-4) exhibits the opposite. The gene composition of the iModulons is shown in Supplementary Table 4. The outliers of the replicates for an ETS variant are reflections of the differences in their genotypes (Supplementary Fig. 2, Supplementary Data 1). **e** Estimation of the number of protons required for the phosphorylation of ADP by ATP synthase (proton-to-ATP ratio) using the ME-model and the experimental data. Individual values of four independently evolved replicates of each ETS variant have been shown on the plot and corresponding median and range of values have been presented. Source Data available in Supplementary Table 3 and provided RNA-seq data.

flux through the coupled metabolic pathways. This balance is governed by the cost of protein synthesis.

Remarkably, the overall proteome allocation to the ATS is similar in the evolved variants and generates the same amount of ATP, enabling them to achieve the same growth rate. The different ways in which the ATS is balanced underlies its plasticity and represents a demonstration of the key systems biology concept of alternate optimal states. These alternate states have a different combination of proton pumping efficiency, complementary metabolic rewiring achieved through tradeoffs in the composition of the transcriptome, and concomitant efficiency of proteome allocation, but enable the same overall cellular function. The cytoplasmic-periplasmic adaptive nexus that the ATS represents thus illustrates the deep plasticity inherent in achieving balanced energetic systems to match metabolic needs in different environmental niches.

## Methods

**Examining PRECISE 2.0 for expression levels of respiratory enzymes**. PRECISE 2.0 is a compendium of high-quality RNA-seq for *E. coli* K-12[6]. It contains 815 RNA-seq datasets of samples with different genetic changes or varied growth conditions. We examined the expression of respiratory dehydrogenases and reductases in the entire dataset. For intelligible purposes, we plotted the expression levels in samples that are directly or indirectly associated with energy metabolism. The expression levels shown are the median value across replicates for a sample.

**Strain generation and adaptive laboratory evolution**. *E. coli* K-12 MG1655 (ATCC 700926) was used as the wild-type strain. P1 phage transduction method was used to generate the knockout strains[41], and strains from the Keio collection were used as a donor for the gene knockout cassettes[42]. *u*ETS-1H and *u*ETS-3H

were generated and used for validation purposes in an earlier study[4]. *u*ETS-2H and *u*ETS-4H were generated here and all four ETS variants were evolved for this study.

ALE was performed using 4 independent replicates of each ETS variant. Cultures were serially propagated on M9 minimal medium with 4 g/L glucose at 37 °C and well-mixed for proper aeration using an automated system that passed the cultures to fresh flasks once they had reached an $A_{600}$ of 0.3 (Tecan Sunrise plate reader, equivalent to an $A_{600}$ of ~1 on a traditional spectrophotometer with a 1 cm path length). Cultures were always maintained in excess nutrient conditions assessed by non-tapering exponential growth. The evolution was performed for a sufficient time interval to allow the cells to reach their fitness plateau.

**Prediction of the effect of amino acid substitutions**. The ALE mutation datasets supporting the conclusions of this article is available in the following open-access archive repository: https://doi.org/10.5281/zenodo.5431595. These datasets are also available in the ALEdb database[10].

Mutated DNA sequence data processing was performed using Python 3. The mutations from ALEdb are described according to their experiment, evolution replicate, sample, and technical replicate. Some evolutions include midpoint samples that could inflate the frequency a mutation is observed. Unique ALE mutations were therefore only considered once per ALE. Starting strain mutations and hypermutator samples were filtered out of the ALE experiment mutation datasets according to their publications. Mutation needle plots were generated using the trackViewer R software package[43]. The visualizations for the 3D protein structures were generated using the *NGL* software package[44]. The software implementation of these actions is available in the following open-access archive repository: https://doi.org/10.5281/zenodo.5431595.

Mutation effects were predicted according to multiple methods. Truncations were predicted according to the potential effect of mutations on the function of start codons and their potential to introduce a premature stop codon. The predicted deleterious effects of SNPs were assumed according to significant SIFT (sorting intolerant from tolerant) scores (SIFT score < 0.05)[19]. The predicted structural destabilization effects of SNPs were assumed according to predicted significant $\Delta\Delta G$ scores ($\Delta\Delta G > 2$)[20]. SIFT and $\Delta\Delta G$ scores were acquired from *Mutfunc*[45]. Functional annotations were acquired from *UniProt*[46] and *Mutfunc*.

**Table 1 Reactions that consume or produce ATP and corresponding stoichiometric coefficient.**

| Reaction[a] | ATP coefficient |
|---|---|
| PPK | −1 |
| PPK2 | −1 |
| ATPS4rpp | 1 |
| PFK_2 | −1 |
| HEX1 | −1 |
| PYK | 1 |
| PFK | −1 |
| PPS | −1 |
| PGK | −1 |
| GLGC | −1 |
| PFK_3 | −1 |
| GART | −1 |
| PPAKr | 1 |
| ACCOAL | −1 |
| ACS | −1 |
| ACKr | −1 |
| SUCOAS | −1 |

[a]Reaction IDs with corresponding reactions in the BiGG Database (bigg.ucsd.edu).

**DNA sequencing and RNA sequencing**. A clone from the endpoints of evolved strains was picked for DNA sequencing and RNA sequencing. The strains were grown in an M9 minimal medium supplemented with 4 g/l glucose. Total DNA was sampled from an overnight grown culture and total RNA was sampled from a culture at an $A_{600}$ ~0.6. Nucleic acid isolation, library preparation, and subsequent analysis were performed as previously described[47]. Briefly, genomic DNA was isolated using a Nucleospin Tissue kit including treatment with RNase A. Resequencing libraries were prepared following the manufacturer's protocol using Nextera XT kit. RNA was isolated using the Qiagen RNeasy Mini Kit following suggested protocol. Ribosomal RNA was removed using Illumina Ribo-zero kit and a KAPA Stranded RNA-Seq Kit (Kapa Biosystems KK8401) was used to prepare sequencing libraries. Sequencing was performed on an Illumina HiSeq and/or NextSeq.

**Phenotype characterization**. Phenotype characterization was performed using two independent biological replicates. Samples for the substrate uptake and secretion rate were collected at regular intervals and filtered using a 0.22 µm filter (PVDF, Millipore). The measurements were performed using refractive index detection by HPLC (Agilent 12600 Infinity) with a Bio-Rad Aminex HPX87-H ion exclusion column. The HPLC method was the following: injection volume of 10 µL and 5 mM $H_2SO_4$ mobile phase set to a flow rate and temperature of 0.5 mL/min and 45 °C, respectively. The phenotype dataset was used for the aero-type classification of the strains as described previously[4].

**Metabolic flux mapping and estimation of H⁺/ATP value for ATP synthase**. Flux mapping was done as previously described using a genome-scale model of metabolism and protein expression[48]. The same FoldME model was used for estimating the H⁺/ATP value for ATP synthase within each ETS variant and replicates. The model was constrained with phenotypic data (glucose uptake rate, acetate production rate) and expression data was layered on using the same methods used for the flux mapping[48]. In addition to these constraints, the necessary ETS genes for each variant were knocked out. Proton pumping ratios from 2.5 to 4.5 were sampled by changing the stoichiometry of the ATPS4rpp reaction in the ME-model, and then the proton pumping ratio was optimized so that the model produced a biomass dilution rate that matched the experimentally determined growth rate.

**ATS proteome allocation calculation**. The same FoldME model was used for the proteome allocation calculation as the flux mapping and ATP synthase estimation calculations. The model was constrained with phenotypic data (glucose uptake rate, acetate production rate, growth rate) and expression data was layered on using the same methods used for the flux mapping. Solutions from the fully constrained ME-models were then used for calculating proteome allocation. Total proteome allocation for each strain was calculated as follows:

$$Total\ Proteome\ Allocation = \sum_i mw_i * V_i^{translation}$$

Where $mw_i$ and $V_i^{translation}$ represents the molecular weight and translation flux of the $i$th protein in the model. Total proteome allocated to the ATS was calculated as follows:

$$Proteome\ Allocated\ to\ ATS = \sum_i mw_i * V_i^{translation}$$

where $mw_i$ and $V_i^{translation}$ represents the molecular weight and translation flux of the $i$th protein in the ATS (209 genes total). The list of 209 ATS genes was generated based on Clusters of Orthologous Groups (COG) and Gene Ontology (GO) categories to include as many relevant genes as possible to represent pathways involved in ATP production, then filtered to remove genes that are never expressed in the multiple model simulations. Mass fraction of proteome allocation to the ATS was calculated as a ratio of the two values for each strain.

Calculation of the total ATP produced by the ATS used the same fully constrained ME-model. A list of all metabolic reactions associated with ATS genes was curated. Reactions that consumed or produced ATP were noted and the stoichiometric coefficient associated with ATP was used as a modifier for calculating the total ATP production as follows (Table 1):

$$Total\ ATP\ Production = \sum_i c_i * V_i^{metabolic}$$

where $c_i$ and $V_i^{metabolic}$ represents the ATP stoichiometric coefficient and the metabolic flux of the $i$th ATS associated reaction in the table below.

Total ATP Production/Total Proteome Allocated was calculated as a ratio of the total ATP production to the mass fraction of proteome allocated to the ATS for each strain.

**ATS transcriptome ICA decomposition**. Independent component analysis was performed on an RNA-seq dataset with steps described in[6]. The only genes included in the dataset were those contained in the list of 209 ATS genes. The dataset consisted of all unevolved strains, uETS-1H through 4H, and all evolved replicates eETS-1HA through eETS-4HD. Additionally, the unevolved and evolved wild-type strains were included with the former being used as a reference to center the data. The final and resulting dataset that was used for ICA contained 209 genes by 22 conditions.

**Reporting summary**. Further information on research design is available in the Nature Research Reporting Summary linked to this article.

## Data availability

Resequencing and expression profiling data that support the findings of this study can be accessed from NCBI Sequence Read Archive accession number PRJNA835443 and Gene Expression Omnibus accession number GSE202144 respectively. The PRECISE compendium and all associated data files can be found at https://github.com/SBRG/precise2. Source data for the figures can be found in Supplementary Tables 2, 3 as well as in the uploaded RNA-seq data.

## Code availability

The software scripts supporting the prediction of mutation effects to the encoding of genes described in this article are available in the following open-access archive repository: https://doi.org/10.5281/zenodo.5431595. All the simulations performed in this manuscript can be reproduced using the FoldME model, which is constructed using the COBRApy toolbox version 0.5.11 for constraint-based modeling and its extension for ME-models, COBRAme version 0.0.9, ECOLIme version 0.0.9, and solveME, all publicly available on Github (https://github.com/SBRG/ME-script, https://github.com/SBRG/ecolime, https://github.com/SBRG/solvemepy). Custom code for constraining and solving ME-models can be found at https://github.com/SBRG/ME-script. GraphPad Prism version 9.2.0 was used for generating the plots.

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

## Acknowledgements
This work was funded by the Novo Nordisk Foundation Grant Numbers NNF10CC1016517 and NNF20CC0035580 and National Institutes of Health Grant R01GM057089. We would like to thank Marc Abrams (Systems Biology Research Group, University of California San Diego) for assistance with paper editing.

## Author contributions
A.A., A.F., and B.O.P. designed the study. A.A., C.O., and R.S. performed the experiments. A.A., A.P., K.C., P.P., C.L., and B.O.P. analyzed the data. A.A. and B.O.P. wrote the paper, with input from all co-authors.

## Competing interests
The authors declare no competing interests.
