## [Peer Review File · Nature Communications]

Reviewers' Comments:

Reviewer #1:

Remarks to the Author:

I read the manuscript by Anand et al 'Laboratory evolution of synthetic electron transport system variants reveals a larger metabolic respiratory system and its plasticity' with great interest, but I must confess that the article left me perplexed. I am an expert in bacterial metabolism and very familiar with the respiratory chain, but not an expert in systems biology. I have read the literature references to become intensively familiar with the topic, but in the end I could not summarize in one or two sentences what the take-home message of the manuscript is. I think that the manuscript is written in a very confusing way, always stays on the surface when presenting the results and it is impossible for a non-expert to grasp the context.

It has long been known that *E. coli* is better able to cope with the loss of NDH-I than with the loss of NDH-II (p. 2). The reason is the approximately 8-fold higher turnover of NDH-II, so that the NADH/NAD⁺ balance of the cell can be maintained. However, NDH-II can restore the balance quickly but not completely, as it has a 10-fold lower affinity for NADH.

How can it be that the non-evolved strain -1H carries a mutation from the beginning (p. 2)? This can falsify all other results with this strain. The evidence that the evolved strains carry mutations that allow faster growth is far too superficial (p. 2).

Strains -3H and -4H are described as evolving in the same direction, with -3H apparently inactivating SDH (p. 2), while strain -4H increases flux across SDH (p. 7/8). So, they do not evolve in the same direction. In strain -4H, the gene *yjjX*, an inosine/xanthosine triphosphatase, was inactivated. The connection of nucleotide metabolism with an artificial linear respiratory chain is not explained.

The main text states that the concentration of the main metabolites was determined. In the supplementary information it becomes clear that actually only glucose and acetate were determined, while the other substances were below the detection limit. I find it very daring to simulate a metabolic flux from the concentration of two metabolites.

The calculated proteome allocation completely disregards the fact that the complexes of the respiratory chain carry cofactors that have to be produced and incorporated. Especially for iron-sulphur centers, this represents a major effort.

The definition of the ATS is basically only a compilation of alternative pathways that can be involved in the synthesis of ATP. Their shares in the total ATP synthesis are somewhat different in the evolved strains and the data are approximated quantitatively here, but they are not in themselves new findings.

Furthermore, the manuscript has some conceptual uncertainties:

The bd oxidases generate a proton gradient via a vectorial process; they do not pump scalar protons (p. 2).

The ATP synthase consists of a motor and a rotor (not two motors) connected by the gamma subunit. The difference in the number of c-ring subunits is due to the difference in pmf, a quantity not considered in this paper. The size of the pmf determines the number of c-ring subunits, and thus how many protons are needed for the synthesis of an ATP. The H⁺/ATP stoichiometry of 3.3 and 4 are both based on the experimentally proven rotational mechanism. The number of 10 c-ring subunits is not preferred but is at the lower end of the scale and is so small only in *E. coli* (p. 10).

Fig. 1A is a misrepresentation of the desired state of affairs, as it implies that, for example, electrons are converted to protons at arbitrary locations across the membrane.

Reviewer #2:

Remarks to the Author:

The manuscript by Anand and co-workers describes a systems biology analysis of four lineages of *Escherichia coli* engineered to restrict the number of protons pumped per electron processed through the electron transport chain. The strains were engineered by deleting combinations of dehydrogenases and oxidases that result in strains capable of pumping 1, 2, 3, or 4 protons per electron donated from NADH. The strains were then adaptively evolved, interestingly reaching identical growth rates after several generations. The strains were sequenced, mutations were identified, RNA was isolated for transcriptome sequencing, and fluxes were estimated by a genome-scale metabolism and expression (ME) model constrained by uptake and excretion measurements. The analysis showed that each strain altered its glucose catabolism and ETS expression levels in response to altered PMF yield per electron. The analysis highlights the available metabolic plasticity in *E. coli* metabolism. Overall, this is a very interesting story, but its current presentation feels terse – lacking full descriptions of methods, connections to prior studies, and complete explanations of conclusions. One would need to read several of the references and be intimately familiar with this group's prior work to understand the conclusions being drawn. In my opinion adding back some of the text that was likely cut to meet other Nature journal limits would make the article more accessible to a general audience. Below are some suggestions for ways to improve the presentation.

1. A figure describing the locations of the full mutation lists for each lineage (picturing genome map with marks for each mutation type) would help clarify the picture of how mutagenized the evolved strains are. Or at least a statement speaking generally about the number of rounds of ALE used, the number of mutations acquired, and how this compares to other ALE studies done in the group. Additionally, an explanation of why the starting strain was chosen would help explain the set of mutations in each of the unevolved strains.
2. At first, I did not see a mutation in table S1 that supported this statement: "The growth improvement of uETS-2H was supported by point mutations in the β' subunit of RNA polymerase." Then realized the statement was referring to the evolved strains picking up this mutation. Please consider revising the statement.
3. The description of mutations in *sdhA* and *yijX* sound like functional deletion is all that is needed. Was this confirmed with deletions of these enzymes? Are these sufficient to restore the improve growth phenotypes seen in the evolved strains?
4. Please replace figure 1 a, with something more descriptive of the four paths, perhaps something like figure 1 of <https://www.pnas.org/content/108/42/17320>.
5. Please provide code files so others can repeat the computational analysis. A more detailed explanation of the ME models would also be helpful.
6. the definition of aerotype on page 8 and how the bounds in figure 3a were calculated were not clearly described. Ref 4 is provided, but a little more information summarizing that study would be appreciated.

Reviewer #3:

Remarks to the Author:

I am Jeremy Wideman, assistant professor in the Center for Mechanisms of Evolution at Arizona State University. I am a proponent of open review and see it as a discussion between scholars. I have extensive experience in eukaryotic and mitochondrial cell biology and evolution. I believe my general knowledge and understanding of the subject matter is adequate to review this paper. I am not a prokaryote expert and some of the difficulties in reviewing this paper stem from this shortcoming. however, I believe this paper is valuable, and should be read across disciplines. Many of my suggestions will hopefully result in a paper that is more accessible to the broad readership that this paper deserves.

Using an elegant KO strategy and an lab evolution approach Anand et al. demonstrate the tradeoff between energetic efficiency and the constrains of a cellular system, in this case the various electron transport paths in *E. coli*. The authors show that regardless of the number of protons pumped per electron transported a similar growth rate is can be achieved by any of 4 different

electron transport paths. The authors determine why this is the case using a combination of transcriptomic and fluxomic(?) methods and use a modeling approach to show that the net ATP per proton pumped is equivalent across all strains.

The data, I believe, are sound, the figures are quite nice, and general message of the paper is compelling.

However, the paper is too short and is lacking in detail and consistency in language that would make it more readable for a general audience. Thus, prior to publication, the paper needs to be written to take advantage of the 5000 word limit in Nature Comms.

Major points:

1. The abstract is impenetrable and needs to be rewritten. The terms aero-type, ETS-4H and aertype system (ATS) are all introduced without sufficient explanation.

I wrote the following so I could make sense of the paper. It may not be entirely correct, but it helped me understand the paper better (I hope).

"Electron transport systems are required to pump protons across a membrane to provide an electrochemical gradient to produce ATP via ATP synthase.

The simplest and most thermodynamically efficient ETSs consist of two enzymes, an NADH:quinone oxidoreductase and a dioxygen reductase, which facilitate the shuttling of electrons from NADH to O₂.

However, evolution has produced variations in ETSs which modulate the overall energetic efficiency of the system even within the same organism.

What tradeoffs exist between these variations remains undetermined.

In order to explore what tradeoffs exist between different ETSs, the authors..."

2. The nomenclature of NDHs are confusing and not consistent throughout the text and the figures. Perhaps it would be simplest to somehow highlight these as pumping (NDH-1) or non-pumping (NDH-2) throughout the text. It would be useful to highlight in each figure the operational respiratory enzymes as outlined in the table of Fig1b. For example, in Fig 2C the flux diagrams would be made simpler if the NDH functional strains (3H and 4H) were labelled as such. Similarly, the CYO and CBD nomenclature and the importance of these differences in electrogenicity is unclear, as is the mention of bd_I and bd-II oxidases.

3. Please describe Figure 2C more thoroughly in text. It looks to me like ETS3H and ETS4H have nearly reciprocally active Krebs cycles with Succ to Fum and akg to Succo having high flux in ETS 4H whereas fum to akg and succo to succ in ETS 3H. This seems very interesting and worth commenting on. However, in text you mention that 4H had a lower flux increase through complex II than 1H or 2H, but this is not reflected in Figure 2C. Perhaps I am misunderstanding something - please clarify.

4. Please clarify what is meant by ETS-4H having more versatility in its adaptation (all strains accumulated mutations in the same gene YjjX, so in what way are there more evolutionary paths?)

5. Some of the supplementary materials (e.g., Fig S3) need to be brought into the main text and better explained. Further, the term iModulons is unfamiliar to me. Is this a new name? Perhaps a new word isn't necessary for this. Figure S4 needs a legitimate figure legend. Actually all the supplementary figures need much more thorough figure legends.

6. P10, first paragraph, last sentence - The authors state that 10 is the preferred number of c-subunits in the c-ring of E. coli. It is unclear to me if any other number has been seen in a structure for E. coli. This is a matter of personal interest, but I also want to ensure that the facts are clear. In the papers cited, it is not clear to me that the authors actually determined the number of c subunits in a functional c-ring as anything other than 10. One paper mentions that c subunit stoichiometry changes when the complex is immunoprecipitated, however, this could indicate that the complex is less stable under certain conditions. I would recommend that this

sentence be removed as there is no clear evidence that the c-ring stoichiometry changes within E. coli. However, this does remain a hopeful possibility.

In sum, with minor rewriting this paper will be suitable for publication in Nature Comms.

April 02, 2022

MS: NCOMMS-21-45711-T

Dear Editor,

We are excited to submit the revised version of our manuscript: *Laboratory evolution of synthetic electron transport system variants reveals a larger metabolic respiratory system and its plasticity*. We would like to thank the reviewers for their valuable manuscript review. We have thoroughly examined the remarks given by the reviewers, and we are presenting our response to each comment in this letter.

Reviewer's Comments: **Black Text**

Authors' Response: **Blue Text**

Reviewer #1:

I read the manuscript by Anand et al 'Laboratory evolution of synthetic electron transport system variants reveals a larger metabolic respiratory system and its plasticity' with great interest, but I must confess that the article left me perplexed. I am an expert in bacterial metabolism and very familiar with the respiratory chain, but not an expert in systems biology. I have read the literature references to become intensively familiar with the topic, but in the end, I could not summarize in one or two sentences what the take-home message of the manuscript is. I think that the manuscript is written in a very confusing way, always stays on the surface when presenting the results and it is impossible for a non-expert to grasp the context.

We took inputs from several non-systems biologists and have worked on improving the presentation of the manuscript accordingly.

It has long been known that E. coli is better able to cope with the loss of NDH-I than with the loss of NDH-II (p. 2). The reason is the approximately 8-fold higher turnover of NDH-II so that the NADH/NAD⁺ balance of the cell can be maintained. However, NDH-II can restore the balance quickly but not completely, as it has a 10-fold lower affinity for NADH.

Yes, this is an interesting tradeoff. We have talked about this aspect in the first submission (page 7, second para).

How can it be that the non-evolved strain -1H carries a mutation from the beginning (p. 2)? This can falsify all other results with this strain. The evidence that the evolved strains carry mutations that allow faster growth is far too superficial (p. 2).

We always sequence the genome of every starting strain. The *pyrE-rph* intergenic mutation is fairly-characterized; it relieves a defect in pyrimidine biosynthesis caused by a 1-bp deletion in the *rph-pyrE* operon. This mutation is common in *E. coli* K12 MG1655 when the strain adapts to minimal media. We believe this mutation was acquired during the gene knockout efforts as the strain passed through several flasks during the protocol. We had a similar observation in one of our previous projects (Table S1 of PMID: 31767748). We have presented our data accordingly and have ensured that nowhere in the manuscript this observation affects the interpretation of our results. As for other mutations, the presence of the same/similar gene mutation in all four independently evolved replicates of each strain suggests causality. Also, we are working on a detailed manuscript on the mechanistic aspects of these mutations.

Strains -3H and -4H are described as evolving in the same direction, with -3H apparently inactivating SDH (p. 2), while strain -4H increases flux across SDH (p. 7/8). So, they do not evolve in the same direction. In strain -4H, the gene *yjjX*, an inosine/xanthosine triphosphatase, was inactivated. The connection of nucleotide metabolism with an artificial linear respiratory chain is not explained.

ETS-3H and ETS-4H have several distinctive features. While their glycolytic flux appears similar, they do have differences in flux across SDH. So, yes, they do not evolve in the same direction in an absolute sense. However compared to ETS-1H and 2H, both ETS-3H and 4H have smaller flux through SDH.

Zheng J. *et al.* (PMID: 16216582) have elaborated on the potential involvement of YjjX in mitigating oxidative stress by preventing the accumulation of altered nucleotides. Another interesting observation was the pulling down of the elongation factor Tu with YjjX, likely to halt protein translation. ETS-4H has the most efficient ETS which in turn may cause elevated ROS production requiring YjjX mediated ROS mitigation. We believe that the evolved ETS-4H (eETS-4H A-D) are mutating the *yjjX* gene to prevent translational halting which will allow them to achieve a higher growth rate. Since these are currently speculative and there is an ongoing study in this direction, we avoided an elaborate discussion in this direction. But we have now detailed this part of the manuscript to guide readers toward a potential explanation.

The main text states that the concentration of the main metabolites was determined. In the supplementary information, it becomes clear that actually only glucose and acetate were determined, while the other substances were below the detection limit. I find it very daring to simulate a metabolic flux from the concentration of two metabolites.

Our HPLC protocol allows us to quantitate all the metabolites that we listed in the manuscript. However, as you rightly pointed out succinate, lactate, formate, ethanol, and pyruvate were below the detection limit. We have removed the list of metabolites from the main text to avoid any over-representation. However, since we are also constraining the model with measured growth rates and expression data, we can be more confident in the simulated metabolic fluxes as opposed to just using the HPLC data.

The calculated proteome allocation completely disregards the fact that the complexes of the respiratory chain carry cofactors that have to be produced and incorporated. Especially for iron-sulphur centers, this represents a major effort.

The reviewer brings up a fundamental issue, namely what are the boundaries of the definition of the ATS. Clearly, the proteome needs support functions to maintain its integrity. These requirements may differ between aerotypes and growth conditions. We have found that support functions, like iron-sulfur centers, represent a small fraction of the proteome. For instance, the ics system is on the order of 0.3%. This number varies a little between the conditions considered in this study, and thus has a small effect on the computation of the overall proteome allocation to the ATS as represented in Figure 3B.

This issue however is deserving more detailed attention. We will point out that we have recently computed the macro and micro-nutrient requirements for growth and how they are reproduced using the ME models (PMID: 34161321). The basic results are that the computation of macronutrient requirements (for growth and energy consumption) is robust, but the micronutrient requirements can be quite intricate and condition-dependent.

The definition of the ATS is basically only a compilation of alternative pathways that can be involved in the synthesis of ATP. Their shares in the total ATP synthesis are somewhat different in the evolved strains and the data are approximated quantitatively here, but they are not in themselves new findings.

We are defining ATS as a combined system, consisting of known metabolic locations of ATP production along with the ETS, which has buffering potential to support a similar ATP supply. We agree with your assessment and therefore we have not claimed it as a new finding.

In systems biology, networks are reconstructed based on existing knowledge. The dynamic/functional state of the system under different conditions is the hallmark of systems biology. This paper focuses on the latter.

Furthermore, the manuscript has some conceptual uncertainties:

The bd oxidases generate a proton gradient via a vectorial process; they do not pump scalar protons (p. 2).

Corrected

The ATP synthase consists of a motor and a rotor (not two motors) connected by the gamma subunit. The difference in the number of c-ring subunits is due to the difference in pmf, a quantity not considered in this paper. The size of the pmf determines the number of c-ring subunits, and thus how many protons are needed for the synthesis of an ATP. The H⁺/ATP stoichiometry of 3.3 and 4 are both based on the experimentally proven rotational mechanism.

The number of 10 c-ring subunits is not preferred but is at the lower end of the scale and is so small only in *E. coli* (p. 10).

We noticed the use of the term 'motor' for both Fo and F1 complexes (PMID: 27321713, PMID: 21524994). Still, we have replaced the term 'motors' with 'complexes' to avoid any technical error.

We are unaware of the information shared by the reviewer that the difference in the number of c-ring subunits is due to differences in PMF. Please share the relevant reading material and we will update the corresponding section suitably.

The preference statement for ATP synthase is in the context of *E. coli*. We have modified the statement to make it clear.

Fig. 1A is a misrepresentation of the desired state of affairs, as it implies that, for example, electrons are converted to protons at arbitrary locations across the membrane.

Based on this comment and a comment from another reviewer, we have modified this figure.

Reviewer #2:

The manuscript by Anand and co-workers describes a systems biology analysis of four lineages of *Escherichia coli* engineered to restrict the number of protons pumped per electron processed through the electron transport chain. The strains were engineered by deleting combinations of dehydrogenases and oxidases that result in strains capable of pumping 1, 2, 3, or 4 protons per electron donated from NADH. The strains were then adaptively evolved, interestingly reaching identical growth rates after several generations. The strains were sequenced, mutations were identified, RNA was isolated for transcriptome sequencing, and fluxes were estimated by a genome-scale metabolism and expression (ME) model constrained by uptake and excretion measurements. The analysis showed that each strain altered its glucose catabolism and ETS expression levels in response to altered PMF yield per electron. The analysis highlights the available metabolic plasticity in *E. coli* metabolism. Overall, this is a very interesting story, but its current presentation feels terse – lacking full descriptions of methods, connections to prior studies, and complete explanations of conclusions. One would need to read several of the references and be intimately familiar with this group's prior work to understand the conclusions being drawn. In my opinion, adding back some of the text that was likely cut to meet other Nature journal limits would make the article more accessible to a general audience. Below are some suggestions for ways to improve the presentation.

Thank you for your appreciation. We have extensively worked on the presentation of the manuscript to make it self-contained.

1. A figure describing the locations of the full mutation lists for each lineage (picturing genome map with marks for each mutation type) would help clarify the picture of how mutagenized the evolved strains are. Or at least a statement speaking generally about the number of rounds of ALE used, the number of mutations acquired, and how this compares to other ALE studies done in the group. Additionally, an explanation of why the starting strain was chosen would help explain the set of mutations in each of the unevolved strains.

We have added the suggested details in the revised manuscript.

2. At first, I did not see a mutation in table S1 that supported this statement: "The growth improvement of uETS-2H was supported by point mutations in the β' subunit of RNA polymerase." Then realized the statement was referring to the evolved strains picking up this mutation. Please consider revising the statement.

We have revised the sentence to avoid any ambiguity.

3. The description of mutations in *sdhA* and *yijX* sound like functional deletion is all that is needed. Was this confirmed with deletions of these enzymes? Are these sufficient to restore the improve growth phenotypes seen in the evolved strains?

Yes, they do seem to be loss of function mutations. However, in our experience, the assessment of causality is more complex and there is an ongoing project to look into the mechanistic aspects of these mutations.

4. Please replace figure 1 a, with something more descriptive of the four paths, perhaps something like figure 1 of <https://www.pnas.org/content/108/42/17320>.

We have suitably modified the figure.

5. Please provide code files so others can repeat the computational analysis. A more detailed explanation of the ME models would also be helpful.

We have included the codes and also added details to the ME model sections.

6. The definition of aerotype on page 8 and how the bounds in figure 3a were calculated were not clearly described. Ref 4 is provided, but a little more information summarizing that study would be appreciated.

As noticed by you as well, we have a full article on the concept of aerotype. The bounds are approximations based on the sampling simulations as described in the previous work. But as per your suggestion, we have added more details to make this concept clear.

Reviewer #3:

I am Jeremy Wideman, assistant professor in the Center for Mechanisms of Evolution at Arizona State University. I am a proponent of open review and see it as a discussion between scholars. I have extensive experience in eukaryotic and mitochondrial cell biology and evolution. I believe my general knowledge and understanding of the subject matter is adequate to review this paper. I am not a prokaryote expert and some of the difficulties in reviewing this paper stem from this shortcoming. However, I believe this paper is valuable, and should be read across disciplines. Many of my suggestions will hopefully result in a paper that is more accessible to the broad readership that this paper deserves.

Using an elegant KO strategy and an lab evolution approach Anand et al. demonstrate the tradeoff between energetic efficiency and the constraints of a cellular system, in this case the various electron transport paths in *E. coli*. The authors show that regardless of the number of protons pumped per electron transported a similar growth rate can be achieved by any of 4 different electron transport paths. The authors determine why this is the case using a combination of transcriptomic and fluxomic(?) methods and use a modeling approach to show that the net ATP per proton pumped is equivalent across all strains.

The data, I believe, are sound, the figures are quite nice, and the general message of the paper is compelling.

Thank you for your appreciation of our work.

However, the paper is too short and is lacking in detail and consistency in language that would make it more readable for a general audience. Thus, prior to publication, the paper needs to be written to take advantage of the 5000 word limit in Nature Comms.

We have extensively worked on this aspect and we hope you will find the revised version more intelligible.

Major points:

1. The abstract is impenetrable and needs to be rewritten. The terms aero-type, ETS-4H and aerotype system (ATS) are all introduced without sufficient explanation.

I wrote the following so I could make sense of the paper. It may not be entirely correct, but it helped me understand the paper better (I hope).

"Electron transport systems are required to pump protons across a membrane to provide an electrochemical gradient to produce ATP via ATP synthase.

The simplest and most thermodynamically efficient ETSs consist of two enzymes, an NADH:quinone oxidoreductase and a dioxygen reductase, which facilitate the shuttling of electrons from NADH to O₂.

However, evolution has produced variations in ETSs which modulate the overall energetic efficiency of the system even within the same organism.

What tradeoffs exist between these variations remains undetermined.

In order to explore what tradeoffs exist between different ETSs, the authors...""

Thank you for a detailed explanation of your comment. Taking lead from here we have improved this section in the revised manuscript.

2. The nomenclature of NDHs are confusing and not consistent throughout the text and the figures. Perhaps it would be simplest to somehow highlight these as pumping (NDH-1) or non-pumping (NDH-2) throughout the text. It would be useful to highlight in each figure the operational respiratory enzymes as outlined in the table of Fig1b. For example, in Fig 2C the flux diagrams would be made simpler if the NDH functional strains (3H and 4H) were labelled as such.

Similarly, the CYO and CBD nomenclature and the importance of there differences in electrogenicity is unclear, as is the mention of bd_I and bd-II oxidases.

NDH-I and NDH-II have been frequently used in the field and therefore, we wish to continue using the terms (PMID: 31133996, PMID: 24444429). We have improved fig 2C as suggested. We have elaborated the description of cytochrome oxidases as well.

3. Please describe Figure 2C more thoroughly in text. It looks to me like ETS3H and ETS4H have nearly reciprocally active Krebs cycles with Succ to Fum and akg to Succoa having high flux in ETS 4H whereas fum to akg and succoa to succ in ETS 3H. This seems very interesting and worth commenting on. However, in text you mention that 4H had a lower flux increase through complex II than 1H or 2H, but this is not reflected in Figure 2C. Perhaps I am misunderstanding something - please clarify.

Figure 2C is showing flux maps of the evolved ETS variants relative to their unevolved strain, rather than absolute flux maps of evolved strains. Therefore, we should only analyze the status of flux after the evolution of a strain and we should not do a cross-comparison since their baselines differ. We have made it more clear in the revised draft.

Figure: Flux through SDH in all the strains of this project.

4. Please clarify what is meant by ETS-4H having more versatility in its adaptation (all strains accumulated mutations in the same gene YjjX, so in what way are there more evolutionary paths?)

This statement is derived from Supp figure 2 where we observe multiple locations for the optimal biomass yield in two-dimensional space according to the respiratory strategies for ETS-4H. This suggested accessibility to more diverse metabolic states (Fig. Supp. 2).

However, we are finding this analysis out of place and it is creating a disconnect in the narration. Therefore, we are motivated to remove this from the revised manuscript.

5. Some of the supplementary materials (e.g., Fig S3) need to be brought into the main text and better explained. Further, the term iModulons is unfamiliar to me. Is this a new name? Perhaps a new word isn't necessary for this. Figure S4 needs a legitimate figure legend. Actually, all the supplementary figures need much more thorough figure legends.

We have moved some details from the supplementary materials to enrich the content of the main text. We are not very comfortable moving fig S3 to the main text because this schematic has been drawn using data already presented elsewhere in this manuscript. Further, we have revised the figure legends.

iModulon is a relatively new term but has already been described in many articles (PMID: 33529205, PMID: 32710928, PMID: 34777307, PMID: 35306876, PMID: 33172971, and more). It allows us to perform an unbiased analysis of the RNAseq data and that is why we are continuing with this concept in this manuscript as well.

6. P10, first paragraph, last sentence - The authors state that 10 is the preferred number of c-subunits in the c-ring of E. coli. It is unclear to me if any other number has been seen in a structure for E. coli. This is a matter of personal interest, but I also want to ensure that the facts are clear. In the papers cited, it is not clear to me that the authors actually determined the number of c subunits in a functional c-ring as anything other than 10. One paper mentions that c subunit stoichiometry changes when the complex is immunoprecipitated, however, this could indicate that the complex is less stable under certain conditions. I would recommend that this sentence be removed as there is no clear evidence that the c-ring stoichiometry changes within E. coli. However, this does remain a hopeful possibility.

The potential variabilities in the number of c-subunits is an interesting proposition. A group has reported (PMID: 9620972) carbon source-dependent variabilities in the number of c-subunits. Others have also indicated a similar possibility (PMID: 22733773). However, if the reviewer is still concerned about the accuracy or validity of the statement, we will be willing to consider removing the statement from the manuscript.

In sum, with minor rewriting, this paper will be suitable for publication in Nature Comms.

We once again thank the reviewers for their valuable insights and suggestions. This has contributed to strengthening the manuscript.

Reviewers' Comments:

Reviewer #1:

Remarks to the Author:

The revised text and illustrations are much more readable and understandable than the original version of the manuscript. In this form the manuscript could be published, however, two small changes should still be considered:

1. As mentioned in my first review, both bd oxidases exhibit a vectorial mechanism (as also described in cited ref. 7). Accordingly, the 4th sentence in section 3 on page 2 (... generate a proton-motive force (PMF) by a vectorial movement of protons ...) and the legend to Figure 1 (... represent the vectorial mode of PMF generation) should be changed.

2. It is generally accepted that the number of c-ring subunits of the ATP synthase motor varies between species but not within a species. This should be clearly described on page 10, second paragraph (... suggests the H⁺/ATP value to be 3.3 in E. coli, due to ...; ... may vary between different species depending upon any change ...). Finally, reference 36 gives an upper thermodynamic limit for this value, in vivo this value may be undercut but never exceeded. Thus, both values do not contradict each other, but reflect differences in in vivo and in vitro measurements. Accordingly, in Figure 3E, the label of the higher value should read 'Thermodynamic calculation based limit'.

Reviewer #2:

Remarks to the Author:

The authors have satisfied my concerns with the revisions to the new manuscript.

Reviewer #3:

Remarks to the Author:

The authors have addressed all my previous concerns.

May 08, 2022

MS: NCOMMS-21-45711A

Dear Editor,

We are glad that the reviewers' are satisfied with the revised version of our manuscript: *Laboratory evolution of synthetic electron transport system variants reveals a larger metabolic respiratory system and its plasticity*. We would once again like to thank the reviewers for their valuable input. We are presenting our response to the two minor comments of the first reviewer in this letter.

Reviewer's Comments: **Black Text**

Authors' Response: **Blue Text**

Reviewer #1:

1. As mentioned in my first review, both bd oxidases exhibit a vectorial mechanism (as also described in cited ref. 7). Accordingly, the 4th sentence in section 3 on page 2 (... generate a proton-motive force (PMF) by a vectorial movement of protons ...) and the legend to Figure 1 (... represent the vectorial mode of PMF generation) should be changed.

We apologize for this oversight and have made the correction at both the places in the manuscript.

2. It is generally accepted that the number of c-ring subunits of the ATP synthase motor varies between species but not within a species. This should be clearly described on page 10, second paragraph (... suggests the H⁺/ATP value to be 3.3 in E. coli, due to ...; ... may vary between different species depending upon any change ...). Finally, reference 36 gives an upper thermodynamic limit for this value, in vivo this value may be undercut but never exceeded. Thus, both values do not contradict each other but reflect differences in vivo and in vitro measurements. Accordingly, in Figure 3E, the label of the higher value should read 'Thermodynamic calculation based limit'.

We have suitably modified the corresponding sections.